# An Algorithm to Bias-Correct and Transform Arctic SMAP-Derived Skin Salinities into Bulk Surface Salinities

David Trossman [1,2,3,*] and Eric Bayler [4]

1 Global Science & Technology, NOAA/NESDIS Center for Satellite Applications and Research (STAR), Greenbelt, MD 20770, USA
2 Department of Oceanography & Coastal Sciences, Louisiana State University, Baton Rouge, LA 70803, USA
3 Center for Computation & Technology, Louisiana State University, Baton Rouge, LA 70803, USA
4 NOAA/NESDIS Center for Satellite Applications and Research (STAR), College Park, MD 20740, USA; eric.bayler@noaa.gov
* Correspondence: dtrossman@lsu.edu

**Abstract:** An algorithmic approach, based on satellite-derived sea-surface ("skin") salinities (SSS), is proposed to correct for errors in SSS retrievals and convert these skin salinities into comparable in-situ ("bulk") salinities for the top-5 m of the subpolar and Arctic Oceans. In preparation for routine assimilation into operational ocean forecast models, Soil Moisture Active Passive (SMAP) satellite Level-2 SSS observations are transformed using Argo float data from the top-5 m of the ocean to address the mismatch between the skin depth of satellite L-band SSS measurements (~1 cm) and the thickness of top model layers (typically at least 1 m). Separate from the challenge of Argo float availability in most of the subpolar and Arctic Oceans, satellite-derived SSS products for these regions currently are not suitable for assimilation for a myriad of other reasons, including erroneous ancillary air-sea forcing/flux products. In the subpolar and Arctic Oceans, the root-mean-square error (RMSE) between the SMAP SSS product and several in-situ salinity observational data sets for the top-5 m is greater than 1.5 pss (Practical Salinity Scale), which can be larger than their temporal variability. Thus, we train a machine-learning algorithm (called a Generalized Additive Model) on in-situ salinities from the top-5 m and an independent air-sea forcing/flux product to convert the SMAP SSS into bulk-salinities, correct biases, and quantify their standard errors. The RMSE between these corrected bulk-salinities and in-situ measurements is less than 1 pss in open ocean regions. Barring persistently problematic data near coasts and ice-pack edges, the corrected bulk-salinity data are in better agreement with in-situ data than their SMAP SSS equivalent.

**Keywords:** salinity; SMAP; skin-effect; bias; air-sea; Arctic; ocean; machine-learning

## 1. Introduction

In this paper, we present an algorithm to bias-correct and convert sea-surface salinity (SSS) fields from L-band passive microwave satellite retrievals at northern high latitudes into surface salinity fields that can be assimilated by ocean modeling systems. Satellite L-band passive microwave observations (Section 2.1) have demonstrated information potential for nearly global depiction of SSS. Satellite SSS derived from NASA's Aquarius mission and Argo products are generally consistent to about 60°N/S, particularly when compared over coarser resolutions [1]. The collected satellite observations from the European Space Agency (ESA) Soil Moisture and Ocean Salinity (SMOS) mission (greater than 10 years) and National Aeronautics and Space Administration (NASA) Soil Moisture Active Passive (SMAP) mission (greater than 5 years) have been summarized and shown to generally agree with in-situ observations [2]. However, monitoring SSS in the Arctic is more challenging. There are many challenges associated with using SMAP-derived SSS observations to monitor Arctic freshwater changes [3], including its accuracy in the colder waters at such high latitudes and utility to monitor variations in the vicinity of ice and/or

coasts. The sources of these accuracy problems include systematic errors in the ancillary wind fields or the wind roughness model that are used in the surface roughness correction, ancillary sea surface temperature (SST) fields, and ancillary sea-ice products that are used in sea-ice contamination correction, all of which are used in the SSS retrievals. Adding to these challenges, temporally-varying SSS drift-like behavior exists in the SMOS data, which, at least partially, accounts for the inability of SMOS to characterize the annual cycle of SSS in the subpolar North Atlantic Ocean [4]. Because SMAP data suffer from fewer problems than SMOS data in the northern high latitudes and a previous study has achieved an improved surface salinity product based on SMOS data [5], we demonstrate the utility of our algorithmic approach using SMAP data.

One difficulty with assimilating SSS into ocean models is the mismatch between the depth levels that L-band satellites observe (top centimeter of the ocean) and the resolution of the top ocean model layer (typically at least one meter). This mismatch can be seen in the in-situ salinity data collected during multiple observational campaigns (e.g., [6]), including two NASA-sponsored Salinity Processes in the Upper-ocean Regional Study (SPURS) campaigns [7,8]: SPURS-1 in the subtropical North Atlantic Ocean and SPURS-2 in the eastern equatorial Pacific Ocean, with a planned third campaign in the Arctic Ocean. We examine whether differences can be reconciled between the "skin" (satellite-derived from the top centimeter) salinity and "bulk" (in-situ at 1–5 m) salinity at high latitudes using an algorithmic approach. If there is a difference in salinity between the top-centimeter and top-meter, or so, of the ocean due to evaporation, precipitation, runoff, ice melt, or freezing/brine rejection effects, then a correction is needed in order to assimilate the satellite SSS observations. Under evaporation, a theory [9] argues for the existence of a salty, cool, sea surface skin layer; however, this theory was revisited after the creation of an air-sea exchange data set [10]. The latter study found that the cooler and saltier skin layer is always statically unstable, and that cooling controls the tendency to overturn, after which it takes 90 times longer to reestablish the skin salinity than the skin temperature. The skin-effect from this theory depends on several air-sea forcing/flux fields, and SSS retrievals depend on some of the same fields using ancillary data. The availability of in-situ and air-sea forcing/flux data sets is, therefore, crucial to the conversion of skin salinity to bulk salinity for use in ocean data assimilation models. The in-situ data are available from several campaigns at high latitudes (Section 2.2), but remain sparse.

Cold-induced biases in satellite-derived skin SSS observations, however, present a problem at high latitudes for data-assimilating ocean models. Due to the strong dependency of density on salinity in the polar regions, SSS can have a significantly higher impact than SST on constraining the modeled circulation of the Arctic Ocean, notably the waters that overlie the warm, salty Atlantic water mass transiting the Norwegian Sea into the Arctic Ocean. Placing upstream constraints on this Atlantic water can significantly impact on the heat imported to the base of the mixed layer along the shelf-basin slopes in the Eastern Arctic, which subsequently impacts the mixed-layer salinity and sea-ice melting in this region [11,12]. Subsequent sea-ice melt, in turn, can influence the Arctic Ocean's salinity [13], which, when exported to the subpolar North Atlantic Ocean, can have consequences for the Atlantic Meridional Overturning Circulation (AMOC) [14–17]. Several theories have been developed to explain the complicated relationships between the sea-ice state, the ocean's salinity, and the circulation in the Arctic context [18–23]. Thus, if SSS can be better constrained in ocean models, then there is the potential to unravel and understand the relationships between sea ice and ocean properties in the Arctic Ocean.

In this study, our primary objectives are to: (1) assess associated biases in a SMAP-derived SSS product relative to in-situ observations in the top-5 m, (2) characterize the statistics of SMAP-derived SSS observations, and (3) assess whether an algorithm to correct for biases and convert the SMAP-derived (skin) salinities to near-surface (bulk) salinities improves their agreement with in-situ observations, thereby permitting ocean data assimilation systems to exploit northern high-latitude SMAP-derived skin SSS. Currently, SMOS and SMAP observations can be adjusted to correlate with Argo float salinity observations in

the top-5 m; but, because there are no Argo floats in the Arctic Ocean, we need an alternative method to effectively correct for both satellite measurement biases, particularly in colder waters, and skin-induced effects that are inconsistent with the model's top-layer thickness. (We refer to the resulting bulk surface salinity product as "corrected" hereafter.) To achieve this objective, we first identify potential issues with Arctic SSS, characterizing the statistics of SMAP SSS and how those data compare with in-situ observations for 2015 through 2019. We then demonstrate the utility of our algorithm for the corrected bulk surface salinity by comparing the root-mean-square error (RMSE) of the algorithm's estimates, relative to the in-situ observations, with the RMSE of the SMAP SSS product, relative to the same in-situ observations. Finally, we characterize the statistics of the algorithm's corrected bulk salinity product derived from SMAP SSS data. The same exercises can be conducted using a SMOS data product. In demonstrating that our algorithm improves the validity of SMAP data, we note that the algorithm can easily be extended to SMOS data. We highlight differences between the original SMAP product and the corrected bulk salinity product derived from SMAP data.

## 2. Data and Methods

### 2.1. Satellite Data

Satellite passive microwave retrievals of ocean salinity exploit the L-Band (1.41 GHz), with SMOS employing a synthetic aperture interferometer and SMAP using a scanning radiometer. Because of the different instrumentation, they have different strengths and weaknesses. For example, SMOS retrievals are known to be challenged near land and the ice-pack edge due to greater off-viewing-angle sea-ice contamination of the salinity signal [24]. SMOS spatial resolution depends on incident angle, spanning from about 40 km near nadir to about 60 km near 55-degrees incident angle [25], whereas SMAP has a fixed incident angle of 40 degrees, with spatial resolution around 40 km [26]. Higher SSS accuracy can be achieved through spatial averaging. Global coverage from SMOS is approximately 3 days, with suborbital repeats being 23 days and the exact repeat period being 149 days [25]. Global coverage from SMAP is exactly 8-days, with nominal global coverage also every 3 days [26]. The 3-day periods mostly have a single data value for each data set so we choose to average over longer periods for each gridded file. Higher SSS accuracy can be achieved through spatial averaging.

In processing the Level-2 SMAP SSS data [27], we perform the following steps. First, we reduce latitude/longitude coordinates in numerical precision to a single digit after the decimal place. This decreases the required computation time and is inconsequential because a resolution of less than tens of kilometers in the horizontal with the raw Level-2 SSS data are not possible without a downscaling technique. This reduced numerical precision allows us to save memory and effectively bin the data. Next, the SMAP data were gridded at 50 km by 50 km resolution by averaging the values over 8 days (chosen because this is the satellite's repeat interval) and aggregating all of the data over 50 km by 50 km boxes. Then, for each grid point, the 8-day averages of SMAP SSS had their trends and seasonal cycles removed over their respective time periods, by using a Generalized Additive Model (GAM—Section 2.4) to fit SSS with a smooth function of time. The residual time series at each grid point had a variance and skewness computed. Lastly, we calculate two statistics from SSS data to examine their spatial distributions relative to marginal ice zones. We compute an anomalously large SSS value statistic at each point on the grid by counting the number of times where the median SSS is exceeded by more than three times the SSS standard deviation. We do not show the same for anomalously small SSS values because the distribution of SSS values tend to be negatively skewed (i.e., the distribution's tail is longer towards smaller SSS values). We also compute mixing length scales according to an established theory [28]. For these mixing length scales, we supplement SMAP data with a gridded product for horizontal spatial gradients of SSS, which comes from the Level-3 daily Earth & Space Research SMOS data.

We compare Level-2 SMAP data with in-situ data, described in the following subsection, from the top-5 m north of 55°N. We do this by searching for data points from SMAP that were within 50 km and 3.5 days of each in-situ observation's horizontal location and sampled time. These selected points, spatio-temporally local to the in-situ data, are used as training data for the statistical model described below.

### 2.2. In-Situ Data

In-situ data provide evidence that skin-salinity values from satellites should be converted into bulk-salinity values in multiple regions around the world. Observations from the SPURS-2 campaign demonstrate that precipitation (or other freshwater flux) into the sea surface have an influence on the skin salinity within the top half-meter of the water column, finer than the vertical resolution of most ocean models. Whether this skin-to-bulk salinity conversion is necessary at high latitudes is currently unknown. However, an additional issue that may be more important at high-latitudes is the error associated with retrieval algorithms for satellite-derived SSS due to low signal-to-noise ratios in cold brightness temperature environments [5].

In order to determine whether the skin-effect and/or biases in satellite-derived high-latitude SSS need to be corrected, we need to use in-situ data in the top-5 m of the water column in subpolar and Arctic Ocean locations. We make use of multiple in-situ data sets, including the salinity and pressure observations from Saildrone [29], Oceans Melting Greenland (OMG) [30], ship-based CTD hydrographic transects, and NOAA's National Centers for Environmental Information (NCEI) Surface Underway Marine Database (SUMD; "Underway" hereafter). The Saildrone sent to the Arctic in 2019 is a wind-powered, unmanned surface water vehicle. The data the Saildrone collect are transmitted via satellite and are available to both researchers and the public. The Underway data comprises uniformly, quality-controlled in-situ sea-surface measurements from thermosalinographs, involving more than 450 ships and unmanned surface vehicles. These data are so extensive that, even when we include all data sets available over the length of the SMOS Arctic time series, the number of data points in the Underway database are orders of magnitude larger than any other data sets used here. The OMG data comprise both CTD and Airborne eXpendable CTD (AXCTD) (CTD probes dropped from aircraft) data collected during the summer months, 2015 to the present, with about 250 probes being dropped each year. Ninety-two ship-based CTD hydrographic transect data sets are used here. The OMG and ship-based CTD hydrographic transect data are subsampled such that we only use data within 5 m of the sea surface.

### 2.3. OAFlux Air-Sea Forcing/Flux Data

We supplement the in-situ salinity data with air-sea forcing/flux fields from the OAFlux product [31,32]. To get the wind stress, data from six Special Sensor Microwave/Imager (SSM/I) sensors, two Special Sensor Microwave Imager/Sounder (SSMIS) sensors, Advanced Microwave Scanning Radiometer for EOS (AMSR-E), WindSat, QuikSCAT, and Advanced Scatterometer (ASCAT-A) were used [33]. The footprint resolution various across the SSM/I sensors is finer with higher frequencies (along × cross-track): 69 km × 43 km at 19 GHz, 50 km × 40 km at 22 GHz, 37 km × 28 km at 37 GHz, and 15 km × 13 km at 85 GHz. One-hundred twenty-six buoy time series were used to calibrate different SSM/I sensors due to known issues with drift. The footprint resolution of the conically scanning SSMIS varies from 14 km × 13 km at 183 GHz to 70 km × 42 km at 19 GHz; for ASMR-E varies from 75 km × 43 km at 6.9 GHz to 6 km × 4 km at 89 GHz; and for WindSat is 40 km × 60 km at 6.8 GHz, 25 km × 38 km at 10.7 GHz, 15 km × 13 km at 18.7 GHz, 12 km × 20 km at 23.8 GHz, and 8 km × 13 km at 37 GHz. The elliptical footprint size of the antenna for QuikSCAT is about 24 km × 31 km at inner beam. For ASCAT, an operational product at spatial resolutions of 25–34 or 50 km can be generated on a nodal grid of 12.5 or 25 km. Rain-contaminated retrievals of wind from microwave sensors were discarded because of known problems under rainy conditions. Surface winds from the European Centre for

Medium-Range Weather Forecasts (ECMWF) Re-Analysis (ERA) interim project [34] and the Climate Forecast System Reanalysis (CFSR) from the National Centers for Environmental Prediction (NCEP) [35] were used to as background data in the synthesis. Sensible and latent heat fluxes were similarly derived using satellite observations (the advanced microwave sounding unit A (AMSU-A) and the Special Sensor Microwave Imager) and reanalyses where and when satellite observations were not available [36], except surface fluxes were computed from the COARE bulk flux algorithm [37]. Evaporation is directly proportional to the latent heat flux and scaled by the inverse product of the density of sea water and the latent heat of vaporization. To get the surface humidity and temperature fields, brightness temperature observations from four vertically polarized channels at 19, 22, and 37 GHz from SSM/I and SSMIS and 52 GHz from AMSU-A were used and related to buoy observations of surface humidity and temperature at 2–3 m above the sea surface [38,39]. The surface humidity and temperature fields were height-adjusted to 2 m using the COARE algorithm [37]. Sea surface temperatures, derived from the global operational NOAA product at 25 km based on AMSR-E and the advanced very high resolution radiometer (AVHRR) [40], were used as constraints for the synthesis of surface humidity and temperature.

The theory of the least-variance linear statistical estimation [41,42] was the basis for the methodology of the OAFlux objective synthesis, using all of the above data constraints. This approach allows the formulation of a least squares estimator (i.e., the cost function) to include both data from different sources and a priori information. For the optimization of each of the turbulence flux fields, a conjugate-gradient method was used [31]. The 25 km resolution of the OAFlux product was chosen as a compromise between being able to satisfy the cost function and the data coverage.

### 2.4. Generalized Additive Model

We use a machine-learning-based approach to convert the satellite skin salinity observations to bulk near-surface salinity that match the salinities measured with in-situ instruments while accounting for high-latitude retrieval biases. The significance of particular terms in the regression equation used will yield evidence of whether the skin-effect and/or biases need to be corrected. Our algorithm of choice is a Generalized Additive Model (GAM) [43]. This machine-learning-based approach, in particular, has a history rooted in statistical regression techniques (e.g., [44]). Ultimately, predictions are made by using predictors (described below) as inputs, just as other statistical regression-based approaches would do. One primary difference between a general linear-regression technique and a GAM is that the latter aims to achieve a balance between the bias and variance of its predictions through a regularization term. This regularization term prevents the machine-learning method from over-fitting to a particular training data set, permitting the approach to be applied to other data sets for prediction purposes. To guarantee that the machine-learning model does not over-fit to the training data, a cross-validation is applied by excluding some of the observations from the training data set, predicting those data, verifying that those predictions are accurate, and then repeating this procedure for different subsets of the training data set.

Instead of estimating the bulk surface salinity, we use a GAM to estimate the bulk surface salinity bias plus skin effect in the satellite-derived SSS data,

$$\Delta SSS_{bulk} = f_0 + f_1(t) + f_2(\Delta SSS) + f_3(SSS_{skin}) \qquad (1)$$
$$+h(SSS_{skin}, SST, \lambda, w_{inv}, Q_{sens}, Q_{lat}, E, q_{hum}, \Delta SSS),$$

where Table 1 describes what each term means.

**Table 1.** Descriptions of each term in Equation (1).

| Term | Description |
|---|---|
| $f_i(\cdot)$ | Smoother functions for $i = 0 \ldots 3$ |
| $h(\cdot)$ | Tensor product of pairwise variables |
| $SSS_{skin}$ | Satellite-derived SSS from $D_{sat}$ |
| $t$ | Julian day relative to January 1 of 1970 |
| $z$ | Depth of the in-situ observations |
| $\lambda$ | $= 6(1 + (16(Q_{sens} + Q_{lat}(1 + S\beta c_p/(\alpha L_e)$ $+0.99 \times 5.67 \times 10^{-8}(SST + 273.16)^4)g\alpha\rho c_p \nu^3 w_{inv}^4/k^2)^{3/4})^{1/3}$ an empirical coefficient as in [10] |
| $Q_{sens}$ | Sensible heat flux from OAFlux [31] |
| $Q_{lat}$ | Latent heat flux from OAFlux [31] |
| $SST_{bulk}$ | Sea-surface temperature in Celsius from OAFlux [31] |
| $L_e$ | Latent heat of evaporation calculated using TEOS-10 [45] |
| $\alpha$ | Thermal expansion coefficient calculated using TEOS-10 [45] |
| $\beta$ | Haline contraction coefficient calculated using TEOS-10 [45] |
| $c_p$ | Specific heat of seawater calculated using TEOS-10 [45] |
| $\nu$ | $= 1.4 \times 10^{-6}$ is the kinematic viscosity of seawater |
| $p$ | Pressure |
| $k$ | $= 0.5715(1 + 0.003SST_{bulk} -$ $1.025 \times 10^{-5}SST_{bulk}^2 + 6.53 \times 10^{-4}p + 0.00029SSS_{bulk})$ thermal conductivity of seawater [46] (in W m$^{-1}$ K$^{-1}$) |
| $g$ | $= 9.806$ m s$^{-2}$ is the acceleration due to gravity |
| $\tau$ | wind stress from OAFlux [32] |
| $\rho$ | in-situ density calculated using TEOS-10 [45] |
| $w_{inv}$ | $= (\tau/\rho)^{-1/2}$ is a function of the inverse wind stress |
| $E$ | Evaporation from OAFlux [31] |
| $q_{hum}$ | Near-surface humidity from OAFlux [31] |
| $\Delta SSS$ | $= f_c SSS_{skin}\lambda E w_{inv}$ bias correction, with proportionality constant $f_c$ [10]; $f_c$ is determined with the GAM |

The Julian day, $t$, is the most important term to include for the satellite-derived SSS data because it aligns satellite observations with when the in-situ observations were taken. $\Delta SSS$ is important to include in the GAM because it at least partially corrects for the skin effect seen in the satellite data; the remaining terms correct for biases. The correlation between $\Delta SSS$ from the co-located SMAP-derived SSS and the in-situ salinity observations in the top-5 m is 0.37, which is significant to the 95% level. However, the skin-effect correction associated with including $\Delta SSS$ in our algorithm reduces the RMSE by less than 10%. The majority of the decrease in RMSE between our algorithmically-calculated bulk salinities and the in-situ observations in the top-5 m can be explained by the other GAM terms, which are associated with bias-correction. The equivalent correlation for the co-located Barcelona Expert Center SMOS-derived SSS [47,48] is higher (0.46), suggesting that the GAM will be different for different data products.

We derive the corrected bulk surface salinities with the following order of operations. At each location and time, we predict the bulk surface salinity biases, $\Delta SSS_{bulk}(x,y,t)$. We then average these biases over the entire time satellite data period to get $\Delta SSS_{bulk}(x,y) = \overline{\Delta SSS_{bulk}(x,y,t)}$. We then add this temporally-averaged bias correction term to the satellite-derived SSS to get $SSS_{bulk}(x,y,t) = SSS_{sat}(x,y,t) + \Delta SSS_{bulk}(x,y)$. The order of these operations is important because $t$ explains some variability that isn't simply related to the seasonal cycle and/or trend. The RMSE between the BEC SMOS $SSS_{skin}$ and the in-situ data in the top-5 m is larger than that between the SMAP $SSS_{skin}$ and the in-situ data in the top-5 m.

An important, but subtle, detail is that both $w_{inv}$ and $\lambda$ depend upon $SSS_{bulk}$ and we will not know $SSS_{bulk}$ everywhere when using the GAM for prediction. If we assume that we know $SSS_{bulk}$ to calculate $w_{inv}$ and $\lambda$, then our GAM can explain 100% of the deviance (with a RMSE of about 0.04%), but $SSS_{bulk}$ is what we aim to predict. If we assume that we know $SSS_{bulk}$, then we would only be able to calculate $SSS_{bulk}$ at the points where we have in-situ data, so we use $SSS_{skin}$ to calculate $w_{inv}$ and $\lambda$ employing TEOS-10. We then estimate the values of $SSS_{bulk}$ with the GAM. We could then iterate by recalculating $w_{inv}$ and $\lambda$ using the predicted values of $SSS_{bulk}$ and subsequently estimate new values for $SSS_{bulk}$, reducing the RMSE with respect to in-situ data, but the time variability of the resulting $SSS_{bulk}$ is not realistic. Thus, we use a single iteration. It is important to include a minimal number of tensor product terms in $g(\cdot)$ because the data close to the coast have large biases, due to land contamination, making the GAM over-fit to the data, resulting in large bias estimates in most locations outside of the training data.

While ocean state estimate outputs suggest that the difference between sea-surface height and ocean-bottom pressure anomalies could be a good proxy for SSS in many locations within the Arctic [49], operational use of coinciding Level-2 sea-surface height and ocean-bottom pressure data with Level-2 SSS data would be limited. Further, sea-surface heights and ocean-bottom pressures decrease the RMSE of the GAM by less than 0.1%; thus, we use the GAM specified in Equation (1).

## 3. Results

We first assess the biases in the satellite-derived SSS products relative to in-situ observations in the top-5 m. Next, we characterize the statistics (mean, standard deviation, seasonal cycle magnitude, skewness, horizontal gradient trends, large anomaly counts, and mixing lengths) of high-latitude satellite-derived SSS observations. Then we use our algorithm to convert the SMAP-derived (skin) salinities to near-surface (bulk) salinities that can be used for data assimilation and characterize the statistics of the skin-effect and bias-corrected surface salinities. We lastly co-locate in-situ observations and the skin-effect and bias-corrected surface salinities to examine whether our algorithm improves the fidelity of the satellite-derived SSS.

### 3.1. Satellite SSS and In-Situ Salinity Comparisons

We sample the satellite-derived SSS within 50 km and 3.5 days of all publicly available in-situ observations [50] of the top-5 m north of 55°N. The number of match-up observations for SMAP data is fewer than that for SMOS; so, for the SMAP observations, there are less data for training the GAM. For example, there are no Marine Mammals Exploring the Oceans Pole to Pole (MEOP) Conductivity, Temperature, and Depth (CTD) [51,52] observations in the top-5 m within 50 km and 3.5 days of SMAP data (Figure 1). There are very few ship-based CTD hydrography and OMG observations that can be compared with the SMAP data. The number of co-located SMAP-derived SSS data points with in-situ salinity observations in the top-5 m are: 2929 observations with ship-based CTD hydrography, 1,710,428 observations with Saildrone, 8,640,999 observations with Underway, and 3219 observations with OMG. For the available ship-based CTD hydrography and Saildrone match-ups with the SMAP data, their disagreement is smaller than in the comparison between OMG and SMAP data. As shown in Figure 1, the Underway data comparisons

with SMAP data have at least two distinct clusters of salinities, one around 32 pss in the North Pacific Ocean and the other around 35 pss in the North Atlantic Ocean. There may be a third cluster of points in the North Sea at salinities between 26–28 pss in the Underway data but much saltier in the SMAP data.

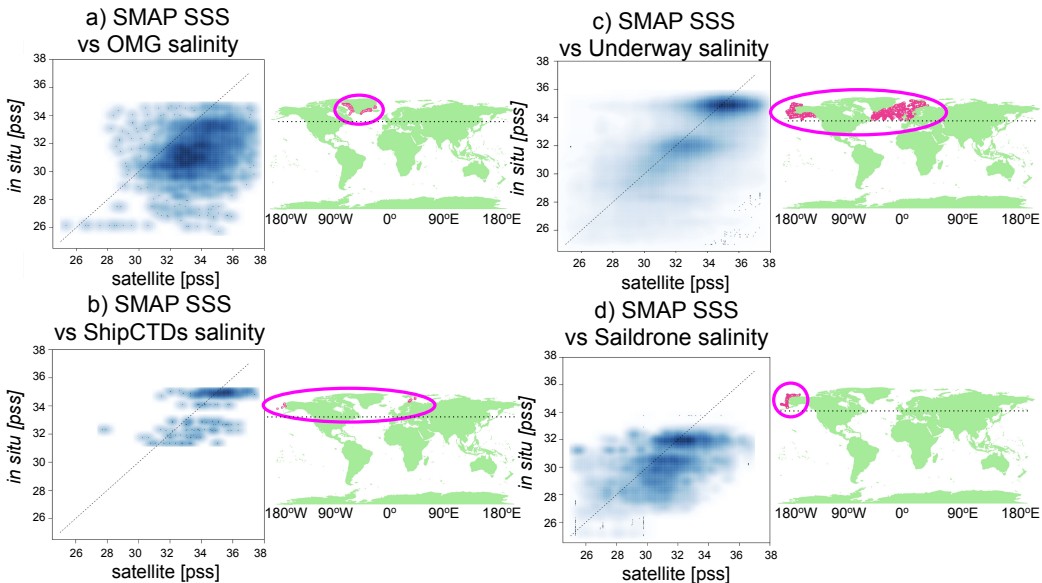

**Figure 1.** SMAP Level-2 skin SSS for April 2015 to December 2020 (abscissa) sampled within 50 km and 3.5 days of all in-situ observations in the top-5 m versus the bulk SSS from in-situ observations in the top-5 m (ordinate) from the Oceans Melting Greenland (OMG— panel **a**), ship-based CTD hydrography (panel **b**), Underway (panel **c**), and Saildrone (panel **d**) campaigns. The darker the shade of blue, the greater the number of points in the scatterplots; outliers are shown with single black dots. Additionally shown next to each scatterplot are the locations where the comparisons between the SMAP SSS product and the in-situ observations are made (pink dots, regions circled); the dashed black line indicates where 55°N is.

We next compare the SMAP data with the aggregated data from all in-situ data campaigns and inspect whether there is any depth-structure to the biases. The Underway data comparisons are very representative of the scatter between the satellite and in-situ data sets (Figure 2a) because they comprise most of the in-situ data. While the differences between the SMAP-derived SSS relative to the in-situ data have many more outliers in the top two meters, there is no statistically distinguishable depth-structure to the differences between the data sets (Figure 2b). The SMAP product has an overall 4.63% (1.54 pss) RMSE relative to the aggregate in-situ data, which is fairly consistent with a similar comparison with in-situ data north of 65°N tabulated in a previous study [53]. This value is relatively small compared to an overall 6.88% (2.29 pss) RMSE between SMOS, different from the product compared in the same previous study, and the aggregate in-situ data. Our values contrast with the ones the previous study reported because of differences in our domains, our versions of the SMAP and SMOS products, and our in-situ data.

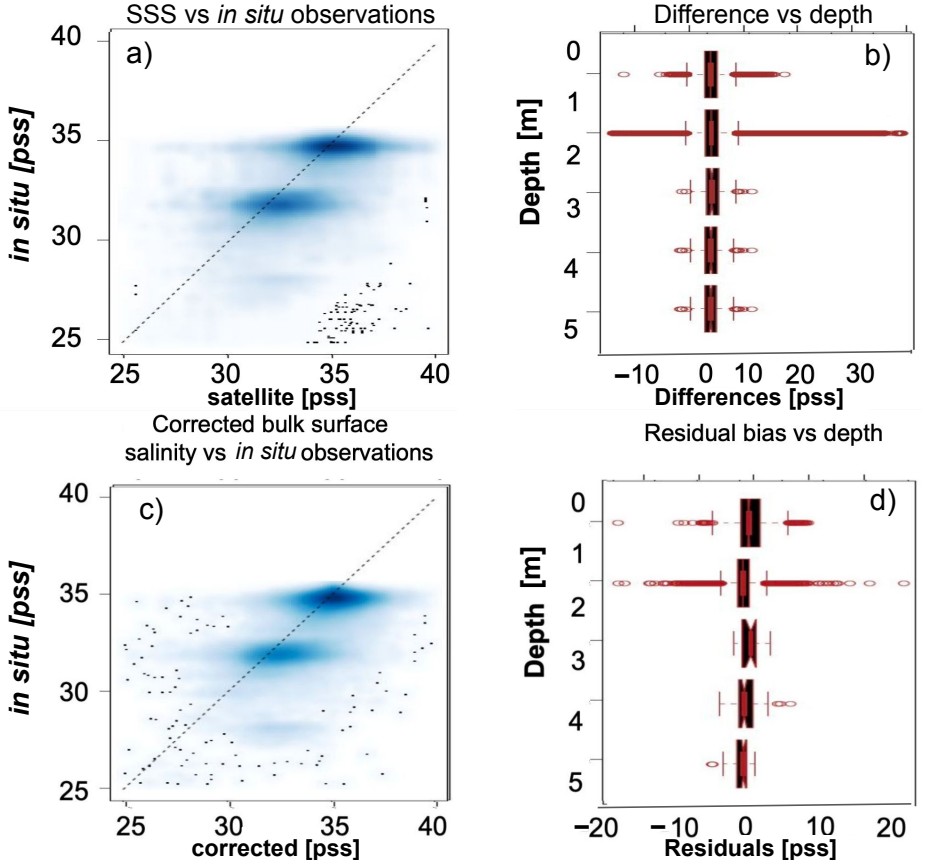

**Figure 2.** Comparison to in-situ salinity observations in the top-5 m (ordinate, panels **a** and **c**): (abscissa, panel **a**) SMAP-derived SSS and (abscissa, panel **c**) corrected bulk surface salinity field with a Generalized Additive Model—GAM, see Equation (1)) using Level-2 SMAP-derived SSS and OAFlux data. The darker the shade of blue, the greater the number of points in the scatterplots; outliers are shown with single black dots. Additionally shown are boxplots of the SSS minus in-situ observations (panel **b**) and corrected bulk surface salinity minus in-situ observations (panel **d**), each as a function of binned depths in the top-5 m. The first depth bin is for 0–1 m, the second depth bin is for 1–2 m, . . . , and the deepest depth bin is for 4–5 m.

After applying our algorithm (Equation (1)) to the satellite and air-sea flux/forcing data sampled at the in-situ data locations and times, we can directly compare the in-situ data with the converted skin-to-bulk salinity data for each satellite data product separately. When trained on 75% of the in-situ data and predicted on the remaining 25%, the RMSE of the skin-to-bulk converted SMAP product relative to the in situ data are reduced to 2.43% (0.81 pss), explaining 71.6% of the deviance. We tested our GAM by training it on 50% of the in-situ data as well, with nearly identical results because the same portion of the phase space with salinity and air-sea forcing/flux factors gets spanned with this training data. We achieved this result by balancing the need to reduce the RMSE relative to random subsets of the in-situ salinity data with the need to not over-fit the GAM. It is possible to achieve a smaller RMSE using more combinations of predictors, but this increases the generalized cross-validation score, suggesting that the algorithm is less capable of estimating bulk surface salinities outside of the in-situ data set. In our final product, there is slightly more variability as a function of depth for our skin-effect and bias-corrected bulk surface salinities (Figure 2d) than for the SMAP-derived SSS (Figure 2b) in comparison with in-situ data over the top-5 m. However, there is no statistically significant depth-structure to the remaining bias in the skin-effect and bias-corrected bulk surface salinities (Figure 2d). As with the comparison of the satellite observations to in-situ observations (Figure 2a), the converted

bulk surface salinity comparisons with in-situ observations (Figure 2c) display the clusters of salinities and generally lie along the one-to-one line. If we, instead, train a single GAM using an indicator function on the skin salinity term for SMOS, versus SMAP data, the RMSE is larger, but comparable (2.50% or 0.83 pss). Although not shown, applying the same Figure 2 analysis to the SMOS converted bulk surface salinities produces generally the same descriptions.

### 3.2. Temporal Statistics of Arctic SSS

Before comparing the in-situ near-surface (bulk) salinities with the satellite-derived (skin) salinities, we present the temporal statistics of the satellite-derived SSS from the SMAP product. When the BEC SMOS data are included, by eye the figures are identical. The SSS is, on average, typically between 33–35 pss, but can be lower to the east of Svalbard (Figure 3a). In regions with lower SSS, the SSS standard deviations (Figure 3b), after detrending and removing the seasonal cycle (Figure 3d), can be as high as 4–5 pss. The standard deviations of SSS tend to get smaller with distance from the perennial, sea-ice-covered regions. The same is true for the SSS skewness (Figure 3c), except the skewness values tend to be negative, indicating a long, relatively fresh SSS tail closer to sea ice and far northern coasts. These large, negative skewnesses could be due to ice melt and/or run-off, unless precipitation events affect SSS more at high northern latitudes than elsewhere. However, these skewnesses are impacted by SSS biases because the skewness is a function of the average SSS. Additionally, the SMAP SSS uncertainties in the Level-2 JPL product, which are estimated errors in the retrievals, are largest in high-latitude regions (Figure 4a). At high northern latitudes, these uncertainties reach 1.5 pss, with standard deviations and seasonal cycle magnitudes at about 0.5 pss (Figure 4b,d). The SMAP SSS uncertainty skewness is most negative in regions affected by ice melt (Figure 4c). While the SMAP SSS uncertainties are smaller than their biases in many locations in the high-latitude oceans, it is likely that the SMAP SSS uncertainties are too small to represent the true uncertainties in high-latitude regions. Our algorithm quantifies the functional uncertainty associated with our model specification, which are standard errors from the GAM, can then be added to the SMAP SSS uncertainties.

Before presenting the bulk surface salinities after conversion and some bias correction, we present an apparent relationship between SSS and sea-ice melting/refreezing to explain the spatial patterns in SSS statistics (Figure 3). Greater temporal fluctuations in SSS near sea ice (Figure 3b–d) can be explained by retreating sea ice leaving relatively fresh water behind as well as by more frequent absence of sea-ice cover resulting from greater salinity values, which have a colder freezing temperature. The seasonal cycle of SSS is largest near the perennial sea-ice edges (Figure 3d), but that has been removed to calculate the standard deviation and skewness. The trend in SSS is a mixture of increasing and decreasing salinity, with no large-scale pattern trend that is significant to the 95% level (not shown). However, anomalously large SSS, found by counting the number of 8-day averages where the average SSS is exceeded by more than three times the SSS standard deviation (see Section 2.1), align close with the marginal ice zones (Figure 5a). Further, although surface forcing dominates eddy stirring, theoretical estimates [28] suggest that the regions with statistically significant horizontal SSS gradients or anomalously high SSS values always occur where the mixing lengths are small (Figure 5b). The fact that mixing lengths are smaller in marginal ice zones is consistent with previously published theory [54]. These results suggest that the biases in satellite-derived SSS in marginal ice zones are not random and may even provide valuable constraints on ocean-sea ice data assimilation systems; this further motivates our skin effect and bias-correction procedure.

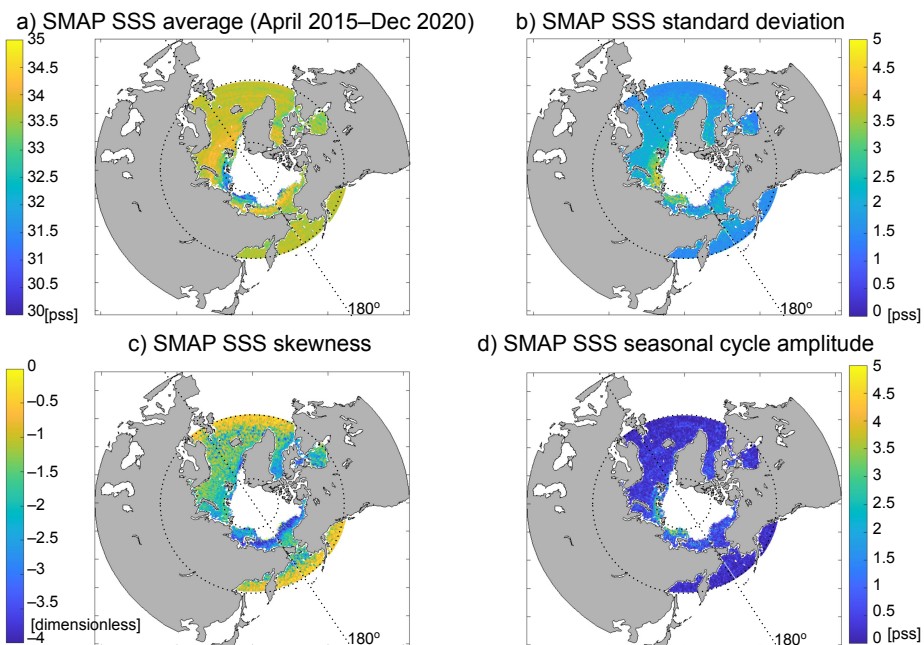

**Figure 3.** Statistics for SMAP Level-2 sea-surface salinity (SSS) product for the period April 2015 to December 2020: (**a**) SSS average, (**b**) SSS standard deviation, (**c**) SSS skewness, and (**d**) SSS seasonal cycle amplitude. The standard deviation and skewness are computed after the removal of the seasonal cycle and trend. The maps synthesize the SMAP data without interpolation, but average all data over each nearest 50 km by 50 km grid point and over each 8-day time period. Overlaid on top are geographical coordinates indicating where the 0° and 180° meridians, as well as the 55°N and 80°N latitudes are.

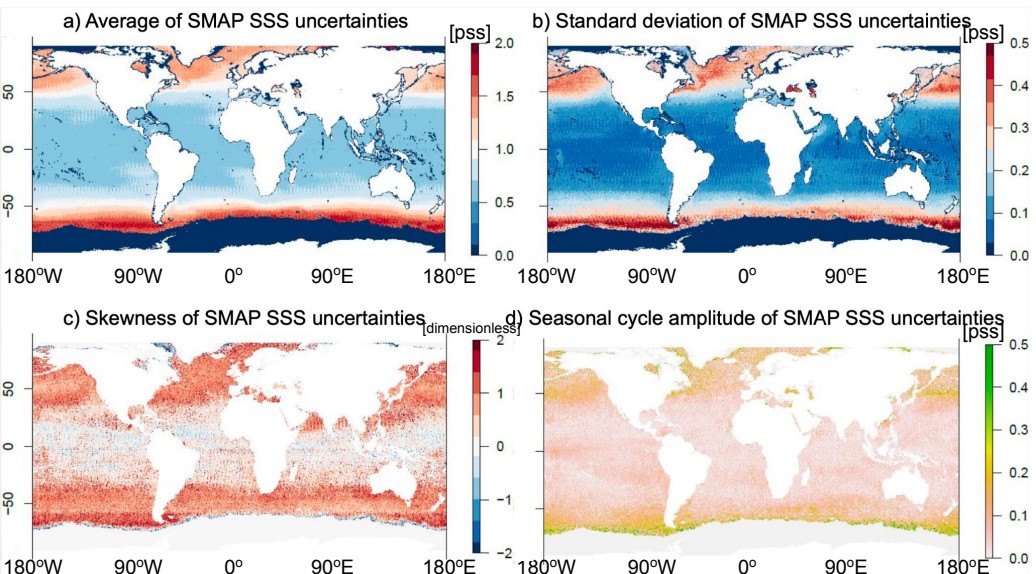

**Figure 4.** Statistics for SMAP Level-2 sea-surface salinity (SSS) product uncertainty for the period April 2015 to December 2020: (**a**) SSS uncertainty average, (**b**) SSS uncertainty standard deviation, (**c**) SSS uncertainty skewness, and (**d**) SSS uncertainty seasonal cycle amplitude. The standard deviation and skewness are computed after the removal of the seasonal cycle and trend. The maps synthesize the SMAP data without interpolation, but averages all data over each nearest 50 km by 50 km grid point and over each 8-day time period.

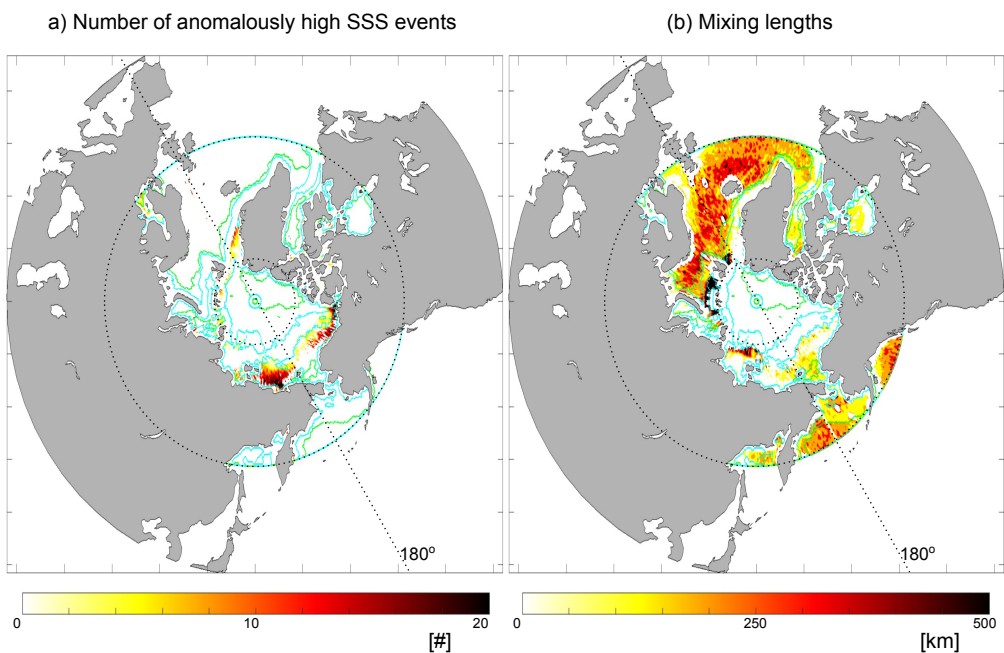

**Figure 5.** High-latitude satellite sea-surface salinity (SSS) anomalies and trends shown in heat-map colors: (**a**) anomalously high SMAP Level-2 SSS product sea-surface salinity (SSS) for the period April 2015 to December 2020, computed by counting the number of 8-day averages where the average SSS is exceeded by more than three times the SSS standard deviation; and (**b**) the mixing length scales calculated as the ratio of the temporal standard deviations from Level-2 SMAP SSS data (April 2015 to December 2020) to the horizontal spatial gradients of SSS from the Level-3 daily Earth & Space Research SMOS product (January 2011 to December 2020). The cyan (green) contours in each panel indicate the minimum (maximum) sea ice extent over all winters between 2015–2019. Overlaid on top are geographical coordinates indicating where the 0° and 180° meridians, as well as the 55°N and 80°N latitudes are.

In Figure 6, we repeat the temporal statistic calculations (Figure 3) using the SMAP data that has been corrected with our algorithmic approach for the skin effect and biases. Figure 6a–d are fairly similar maps to those in Figure 3a–d, but there are some important differences. The average corrected bulk surface salinity values are fresher in the North Pacific, Bering Sea, Chukchi Sea, Davis Strait, Hudson Bay, and coastal Greenland regions and saltier in the subpolar North Atlantic Ocean, Norwegian Sea, and Barents Sea regions (Figures 3a and 6a). The fresher corrected bulk surface salinity values near the Greenland coasts are in better agreement with the in-situ data from the OMG campaign than the satellite-derived SSS values. The corrected bulk surface salinity standard deviations and skewnesses have large magnitudes only for narrow bands near the perennial sea ice and the coasts (Figure 6b,c), as opposed to a larger area over the marginal ice zones (Figure 3b,c). Relative to the satellite-derived SSS seasonal cycle amplitudes (Figure 3d), there are large corrected surface salinity seasonal cycle amplitudes for a greater proportion of the marginal ice zones on the Siberian Shelf (Figure 6d). The algorithm we apply to calculate the corrected bulk surface salinities improves their agreement with in-situ data, but we need additional tests to determine whether the corrected bulk surface salinities are more realistic than the satellite-derived SSS.

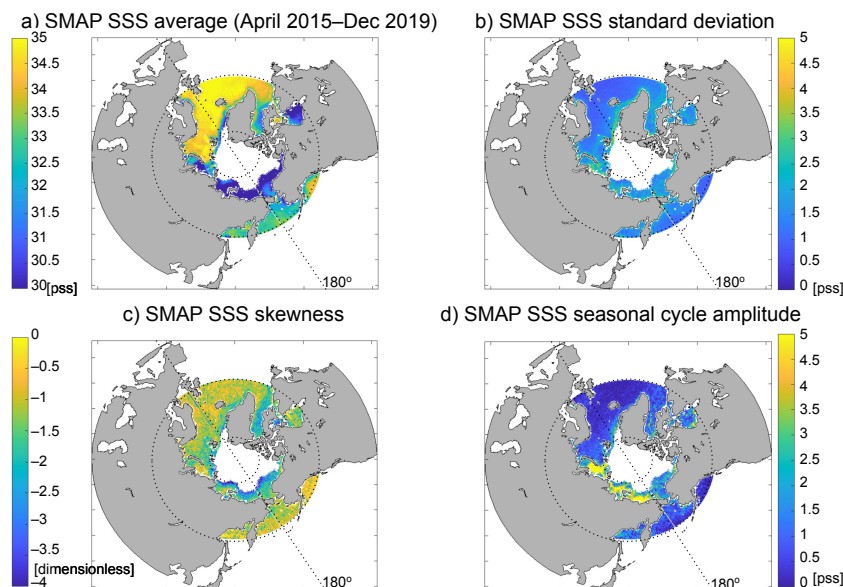

**Figure 6.** Statistics for SMAP Level-2 sea-surface salinity (SSS) product, corrected for the skin effect and bias (April 2015 to December 2019 due to the time range of the available OAFlux data): (**a**) SSS average, (**b**) SSS standard deviation, (**c**) SSS skewness, and (**d**) SSS seasonal cycle amplitude. The standard deviation and skewness are computed after the removal of the seasonal cycle and trend. The product synthesizes the products without interpolation, but averages all data over each nearest 50 km by 50 km grid point and over each 8-day time period, the same as for the statistics shown in Figure 3. Overlaid on top are geographical coordinates indicating where the 0° and 180° meridians, as well as the 55°N and 80°N latitudes are.

One test of how realistic the corrected bulk surface salinities are is to repeat the in-situ data comparisons shown in Figure 1. Because we only correct the time-mean bias at each horizontal location, it is possible that the instantaneous disagreements between the corrected bulk surface salinities and the near-surface in-situ data are about the same or worse; however, the skin-effect and bias corrections do not degrade the accuracy of the salinities relative to the in-situ data (Figure 7). The disagreements between the corrected bulk surface salinity product from SMAP and each of these in-situ data sets are typically less than 2 pss (<1 pss overall RMSE), with negligible bias overall, but disagreements near the coasts, where there is freshening from ice sheet melt, can be much greater. For example, there remains a 2 pss bias in the corrected bulk surface salinity product from SMAP SSS data relative to the OMG data (not shown). These biases are well within the uncertainties associated with the SMAP SSS product (~1.5 pss; Figure 4a) plus with the uncertainties associated with our algorithm (1–2 pss; not shown). After skin-effect and bias-correction, the corrected bulk surface salinities show three distinct clusters of salinities relative to the Underway data: (1) between 26–27 pss, (2) around 32 pss, and (3) around 35 pss. Overall, the corrected bulk surface salinities are improved relative to the Saildrone and Underway in-situ data sets, which comprise the shallowest data relative to any other data sets, but near-coastal satellite data points, where there is potential land contamination, should be removed.

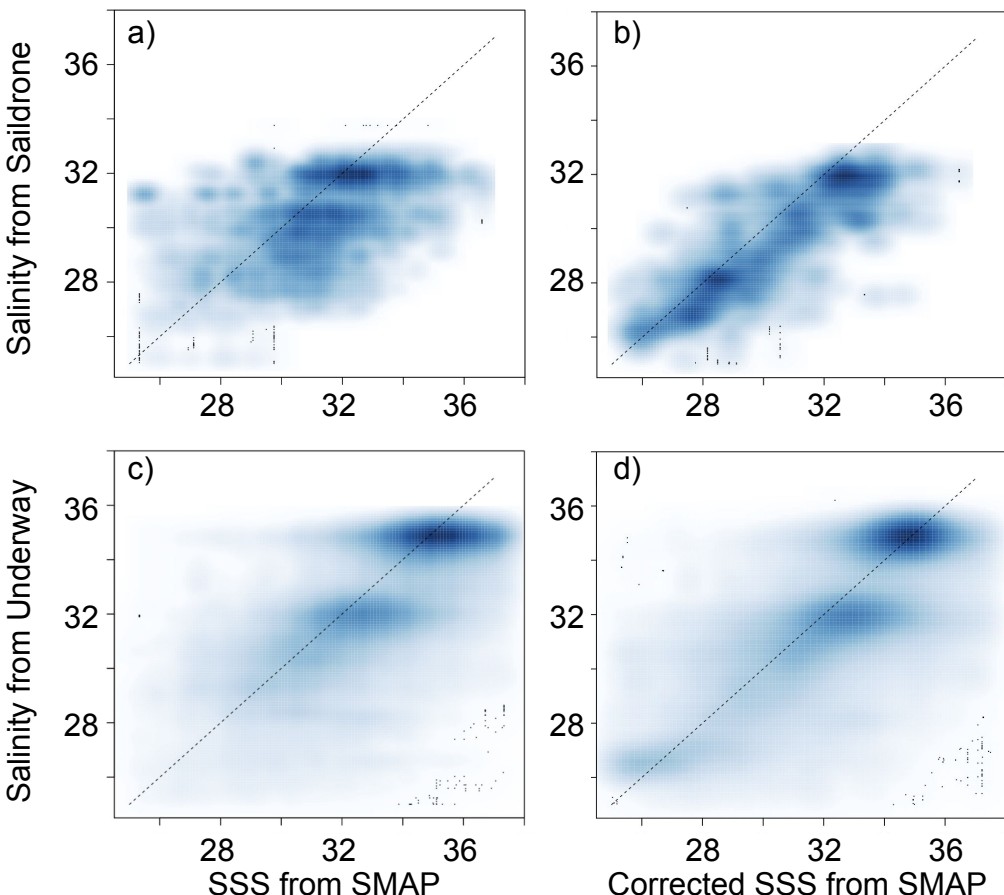

**Figure 7.** Comparison of in-situ and Level-2 SMAP SSS observations (sampled within 50 km and 3.5 days of in-situ bulk surface salinity observations) for the period of April 2015 to December 2019 (abscissa = SMAP, ordinate = in-situ): Level-2 SMAP (panels **a,c**) and skin-effect and bias-corrected Level-2 SMAP (panels **b,d**) data; Saildrone (panels **a,b**) and Underway (panels **c,d**). The darker the shade of blue, the greater the number of points in the scatterplots; outliers are shown with single black dots.

## 4. Discussion

One motivation for developing our algorithm is to convert skin salinities to bulk salinities to allow for data assimilation of satellite-derived salinity data in regions without Argo data, but the more important alteration of the satellite-derived SSS for assimilation purposes is its inaccuracy at high-latitudes. Our algorithm includes terms for multiple air-sea forcing/flux fields, including the wind stress, bulk SST, and the implicit exclusion of regions covered by sea-ice, by making use of the OAFlux product. One interpretation of our results is that these fields provide corrections to the equivalent ancillary fields used in SMAP-derived SSS retrievals. However, an alternative interpretation of our results is that the relevant terms in the Yu (2010) [10] theory and/or additional terms in our algorithm, together, are proxies for inaccuracies in the SMAP-derived SSS retrievals. Each of these interpretations are supported by the facts that: (1) relative to available in situ data, our algorithm reduces the RMSE of the corrected SMAP salinities, and (2) in order to minimize the RMSE, compared with other possible GAM term combinations, our algorithm requires the same terms for both SMAP and SMOS data.

Where they overlap, the spatial distributions of high-latitude surface salinity statistics from our algorithm are consistent with those presented in other observational product comparison studies [4,53,55–57]. Each of these products are within the ~2.0–3.5 pss uncertainties we find in high-latitude regions, after skin-effect and bias-correction. The spatial

patterns are also consistent with previously presented products of surface salinity. Like our product, other Aquarius-based products, SMOS-based products, and the World Ocean Atlas suggest there is more fresh surface water near the Pacific and saltier surface water near the Atlantic [53,56]. Additionally, like our product, these other products suggests there are larger seasonal cycles in surface salinity closer to the Pacific than to the Atlantic. In subpolar regions, the seasonal cycle magnitudes of SSS are small (<1 pss), as shown in previous studies [4,57], especially in comparison with the seasonal cycles both other observational products and our algorithmically-derived product suggest near the Arctic Eurasian and Canadian coasts. The primary differences we find with our algorithmically-derived product in the spatial distribution of the surface salinity statistics are in the relative magnitudes of the standard deviations versus seasonal cycles in these Arctic coastal regions. This is conceivably a result of the temporal resolution of the OAFlux product. However, in these coastal regions and in other locations in the marginal ice zone, a seasonal cycle cannot be accurately estimated due to the seasonal aliasing of the satellite-derived SSS so estimates of the seasonal cycle magnitudes and higher-order statistics that rely on extraction of the seasonal cycle (e.g., standard deviations) are not reliable in these places.

## 5. Conclusions

This study presented a method to convert satellite skin salinity observations to bulk salinity for assimilation into modeling systems. The temporal statistics in a satellite-derived data set of SSS reveal likely influence from sea-ice melt in marginal ice zones. Trends were not detectable over the 5-year period of the data record. Data collected from the SPURS-1 campaign (not shown here) suggested that there could be non-constant structure to salinity profiles, even within the upper-50 centimeters of the water column. Point-by-point comparisons of the satellite SSS with several different sources of in-situ observations in the top-5 m for northern high-latitude regions demonstrated that different geographic regions have different clusters of salinity values and that the satellite-derived data do not agree well enough with the in-situ data for data assimilation purposes. The disagreements between the satellite and in-situ data exceed 1.5 pss, which can be greater than the temporal variability in the satellite data. We presented an algorithm, based on machine learning and trained on the in-situ salinity data and air-sea flux/forcing data, to convert skin-salinities to bulk-salinities. This algorithm for corrected bulk surface salinities cut the disagreement with the in-situ data down by at least half from the comparison between the satellite and in-situ data. The algorithm can reduce the disagreement to a level of less than 1 pss and can produce uncertainties that are simply propagated along with the Level-2 product uncertainties.

The algorithm to convert skin salinities to bulk salinities and correct for biases can be improved and used in multiple applications. First, we can repeat the application of our algorithm with an improved SMAP product after the removal of sea-ice contamination [58] as well as using a SMOS product. We also expect the upcoming SPURS-3 (Salinity and Stratification at the Sea Ice Edge or SASSIE) campaign to enhance the accuracy of our algorithm by making available a greater amount of variance data in the near-surface salinity field. Future air-sea forcing/flux data potentially will be provided in the future by using an observing system called FluxSat that could reduce air-sea flux observational errors by 50% [59]. Our algorithm can be applied to different depth ranges for the in-situ data, as well as in the Antarctic region, depending upon the resolution and domain of the modeling system assimilating the corrected bulk salinities. Both OAFlux and in-situ data would be required for these domains as well, with no additional requirements. The assimilation of bulk salinities can potentially constrain the salinity field at high latitudes, allowing models to evaluate the sensitivies of the surface salinity field to other model processes and parameters. A more realistic surface salinity field in the Arctic could enable better simulation/representation of sea-ice formation/melt, allowing coupled ocean-sea ice-atmosphere models to improve their representation of heat and moisture fluxes. Our algorithm and potential refinements allow for the possibility of these studies and more.

Future observing system evaluation (OSE) studies need to demonstrate improvements in representing upper-ocean hydrography and sea-ice properties to conclude that corrected bulk salinity data have value for assimilation purposes.

**Author Contributions:** Conceptualization, D.T. and E.B.; methodology, D.T.; software, D.T.; validation, D.T.; formal analysis, D.T.; investigation, D.T.; resources, E.B.; data curation, E.B.; writing—original draft preparation, D.T.; writing—review and editing, D.T. and E.B.; visualization, D.T.; supervision, E.B.; project administration, E.B.; funding acquisition, E.B. All authors have read and agreed to the published version of the manuscript.

**Funding:** This research was funded by the NOAA/NESDIS Center for Satellite Applications and Research (STAR).

**Institutional Review Board Statement:** Not applicable.

**Informed Consent Statement:** Not applicable.

**Data Availability Statement:** Sea ice extent: https://nsidc.org/data/G10017/versions/1, accessed on 19 March 2021 (U.S. National Ice Center. 2020. U.S. National Ice Center Daily Marginal Ice Zone Products, Version 1. [northern]. Boulder, Colorado USA. NSIDC: National Snow and Ice Data Center. https://doi.org/10.7265/ggcq-1m67.) Underway: https://www.ncei.noaa.gov/access/surface-underway-marine-database/, accessed on 6 January 2021 (Wang, Zhankun; NOAA National Centers for Environmental Information (2017). Quality-controlled sea surface marine physical, meteorological and other in-situ measurements from the NCEI Surface Underway Marine Database (NCEI-SUMD). [sea surface salinity]. NOAA National Centers for Environmental Information. Dataset at https://www.ncei.noaa.gov/archive/accession/NCEI-SUMD.). OMG: https://podaac.jpl.nasa.gov/dataset/OMG_L2_AXCTD, accessed on 26 February 2021 (OMG. 2019. OMG AXCTD Profiles. Ver. 1. PO.DAAC, CA, USA. Datase at https://doi.org/10.5067/OMGEV-AXCT1) and https://podaac.jpl.nasa.gov/dataset/OMG_L2_CTD, accessed on 26 February 2021 (OMG. 2020. OMG CTD Conductivity Temperature Depth. Ver. 1. PO.DAAC, CA, USA. Dataset at https://doi.org/10.5067/OMGEV-CTDS1). WOCE/CLIVAR/GO-SHIP: https://cchdo.ucsd.edu/search?bbox=-180,55,180,90, accessed on 1 March 2021 ([Cutter, G., Thierry, V., Jeansson, E., Gary, S. F., Lee, C., Ivanov, V., Ashik, I., Schauer, U., Kadko, D., Gobat, J., King, B. A., Holliday, N. P., Olsen, A., MacDonald, A., Mecking, S., Bullister, J. L., Baringer, M. O., Kieke, D., Yashayaev, I., Griffiths, C. R., Skagseth, ø., Fernández Rios, A., Beszczynska-Möller, A., McGrath, G., Read, J. F.]. [2021]. [CTD] data from cruise [33RR20180918, 35A320170715, 58GS20160802, 74EQ20160607, 316N20150906, RUB320150819, 06AQ20150817, 33HQ20150809, 33RO20150525, 58GS20150410, 74JC20140606, 316N20130914, 33RO20130803, 58GS20130717, 06M220130509, 18HU20130507, 740H20130506, 58HJ20120807, 74E320120731, 29AH20120622, 06AQ20120614, 18MF20120601, 45CE20120105, 316N20111002, 06MT20110624, 74E320110511, 18HU20110506, 45CE20110103], [NetCDF]. Accessed from CCHDO [https://cchdo.ucsd.edu/search?bbox=-180,55,180,90]. [n/a].). Saildrone Arctic: https://podaac.jpl.nasa.gov/dataset/SAILDRONE_ARCTIC, accessed on 20 November 2020 (Saildrone. 2020. Saildrone Arctic NOPP-MISST Field Campaign Products. Ver. 1.0. PO.DAAC, CA, USA. Dataset at https://doi.org/10.5067/SDRON-NOPP0). MEOP SEaOS: http://www.meop.net/database/meop-databases/density-of-data.html, accessed on 16 November 2020. OAFlux: https://oaflux.whoi.edu/data-access/, accessed in 16 November 2020. ESR SMOS SSS gradient: https://salinitydata.org/files/data/daily/Salinity/SMOS/locean_debiasedSSS_09days_v5/, accessed on 10 March 2020. JPL Level-2 SMAP SSS: https://podaac.jpl.nasa.gov/dataset/SMAP_JPL_L2B_SSS_CAP_V5, accessed on 15 April 2021 (JPL. 2020. JPL CAP SMAP Sea Surface Salinity Products. Ver. 5.0. PO.DAAC, CA, USA. Dataset at https://doi.org/10.5067/SMP50-2TOCS). The data used to generate our figures are available at https://doi.org/10.5281/zenodo.6353521.

**Acknowledgments:** The authors thank NOAA STAR IT for their support, and the reviewers of this manuscript for their suggestions. The scientific results and conclusions, as well as any views or opinions expressed herein, are those of the authors and do not necessarily reflect those of NOAA or the Department of Commerce.

**Conflicts of Interest:** The authors declare no conflict of interest.

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
