# Peer review of "An Algorithm to Bias-Correct and Transform Arctic SMAP-Derived Skin Salinities into Bulk Surface Salinities"

_remotesensing, doi:10.3390/rs14061418_

Round 1

Reviewer 1 Report

see attached file

Reviewer 2 Report

In the study, the authors presented a method to convert Arctic SMAP-derived skin salinities to bulk salinities for assimilation into modelling systems. To resolve the research task the authors used a Generalized Additive Model. Although the research is interesting and important for the proper estimation of salinity in the Arctic region, I have a few remarks which may improve the manuscript. First of all,  add the discussion section and improve quality of figures. Moreover, format the ms following the instructions of MDPI Remote Sensing. Specific comments are given below.

Page 1, Line 13: write the name of machine-learning algorithm used

Page 1, Line 19: rather than “satellite” present satellite sensor used

Page 1, Line 26: while citing number references following the MDPI instructions. Start with [1].

Page 3, Title of chapter 2: modify the title of chapter. In my view, this chapter also deals with data used. In my view, a map of the study area with geographical names mentioned in the ms could be useful to a potential reader.

Page 5, Lines 194 & 198. Are these correlation coefficients statistically significant at a level of 0.05?

Page 3-6: you use a huge amount of data. What time is necessary to carry out the calculations?

Page 6, Lines 230-235: in my view, objectives of the study should be inserted into the end of Introduction section.

Page 7, Figure 1: improve the quality of Figure 1. First of all, subfigures are really too small. As a result the content is hardly visible. Plots are fuzzy and font size (letters and values near abscissa and ordinate) are too small. Maps are too small and without any geographic information.

Page 8, Figure 2: add units near the X & Y axes

Page 9, Figure 3; Page 10, Figure 4: improve the quality of figures. Add geographical coordinates, units near color bars, increase font size, etc.

Page 11, Figure 5: improve the quality of figure 5

Page 12, Figure 6: improve the quality of figure 6. Add geographical coordinates, units near color bars, increase font size, etc.

Page 13, Figure 7: improve the clearness of figure 7.

Page 13: there is no discussion section at all. Add it please. Among others, discuss your results with earlier publications dealing with the investigated issue.

Page 15: Format references according to MDPI standards.

Reviewer 3 Report

This paper seeks to use a machine learning method to generate a correction for skin-to-bulk sea surface salinity in the Arctic.

I have no knowledge of the machine learning method the paper uses (a "Generalized Additive Model"). Thus I cannot evaluate whether this method is appropriate to the use it is put in this paper, or if it is applied correctly. The paper should also be evaluated by someone with a better understanding of GAMs than me.

I stopped reading this paper on line 316. It is too poorly crafted and filled with mistakes to spend any more time on. As indicated by the volume of comments below, the paper does not meet the quality standards to be expected of work published in Remote Sensing.

As can be seen by some of the comments below, the introduction is confusing as to the purpose of the paper. Is it to develop an improved product for assimilation at high latitudes only, or is it supposed to be useful throughout the ocean? Ditto lines 230-236. The title would suggeest the former, but the discussion here would also suggest the latter.

Olmedo et al. (reference 29 - referred to only in passing in the text) have produced a product especially suited to the Arctic. How does the product produced in this work compare?

21-23. This is where the specificity to the Arctic is lost, in the first sentence of the paper.

26. I think the RS editorial policy is that references are listed in order of appearance in the paper. So the reference on this line should be "[1]". The editor can verify.

37-39. This does not quite make sense. Maybe delete the part on line 39.

83-85. So the intent of this paper is to produce this algorithm for use in high latitudes only? Or is it to produce an algorithm for use in all latitudes?

103-115. Some references are needed in this paragraph.

116-119. This sentence does not make sense. Can the authors please explain better what they did and why.

128-129. I guess if the distribution were Gaussian, we would expect this number to be (trying to remember my basic statistics...) about 0.1% of all observations. If the distribution is negatively skewed (line 130), the number should be much less than that.

139-140. What is a "non-constant profile structure"? Over what depth range? There must have been very few observations during SPURS-1 with SSS<25.

141. I do not recall there being many measurements during SPURS-1 where the 35 cm salinity could be distinguished from the 5 cm value. Which SPURS-1 dataset are the authors talking about? (SPURS-2 is a different story.)

145-147. I do not understand what the authors are saying. And what is the distinction here between high and mid- or low latitudes?

209. I'm not sure about lambda, but w_inv, the inverse of the square root of the wind stress, depends only *very* weakly on SSS through the density in the numerator.

226. I do not understand what the authors are saying. What limits the use of coincident SSH and SSS data?

240-243. Figure 1 says nothing about the MEOP dataset.

243. "very few" Specify a number.

Figure 1. The top panels use a blue shading and dots that are not described or specified (ditto Fig. 2a,c). What does it mean? The bottom panels show the entire globe when just the Arctic is actually needed, making the observation locations (red dots) difficult to see.

249. "There may be a third cluster..." ...at a salinity of...

250. "...all in situ data..." I am trying to figure out what data are actually being used here without much success. Section 2.2 describes a bunch of different programs (OMG, SUMD, saildrone, etc.). Is this different from the data displayed in Figure 1? Maybe this means in situ data that both are and are not matched with SMAP. But these data are actually compared with SMAP, so how does that work?

253-254. Looking at Fig. 2b I disagree. It looks to me like there are many more (especially positive) outliers in the 2 m bin than the others. The medians and quartiles are similar.

255-256. A percentage error is not very descriptive in this context. RMSE in practical salinity units would be much more informative.

260. One wonders how the authors chose this ratio of training to prediction data, and what the consequences would be of a different choice.

267. "no discernible depth-structure" It is hard to tell if this is true from looking at Fig. 2d because of the way the axes are set up. However, it does look as though there actually is some depth structure to the median and quartile values as opposed to Fig. 2b.

Fig. 2b,d. How do the y-axis labels correspond to the binned observations? Maybe the first bin is surface-0.5 m, the second 0.5-1.5, etc.

268. "because the in situ observations are exclusively within the upper-5 meters" What does this prove about the depth structure?

Figure 3. The blue areas in the north must be ones where sea ice contamination severely limits retrievals, so the SSS stats are ill-defined. The figure should indicate where this is the case. Those blue areaes become white in Fig. 3c for some reason.

Figure 3. The STD displayed in Fig. 3b is after removal of the seasonal cycle (line 282). This needs to be stated in the caption. Is this also true for Fig. 4b?

282. "4-5 pss" There are no areas in Figure 3b with values this high.

288. "impacted" how? Please elaborate.

289. "uncertainties" I looked for a definition of unceretainty somewhere in the methods section of the paper with no success. I would also note that the word is used three times in the abstract without having been defined in the paper.

290. "1.5-2 pss" There are no areas in Figure 4a with values this high.

292-293. No map is displayed of bias in Fig. 4, so this statement cannot be verified. What is the difference between uncertainty and bias in this context? Why would one expect uncertainty and bias to be comparable? Why is it notable that bias is larger than uncertainty, i.e. what does that tell us?

298. "by more frequent absence of sea-ice cover" I don't understand what this means or how it could occur near sea ice (line 297).

300. How can a seasonal cycle be determined near sea ice when there is no data there for most of the year?

301-303. A trend is a salinity that increases over time not "salinifying and freshening". I kind of understand what the authors are saying but it is not worded well.

303-304. "Earth & Space Research Level-3 daily product" As far as I can tell this is the first mention of this product. It is not discussed in section 2. Is it a SSS product or SSS gradient?

Figures 5a,b are unreadable.

305. The authors are computing trends in the gradient here. I'm not convinced this is a valid exercise, especially in the Arctic. Neither do I understand what the trend in the gradient has to do with the discussion here.

429. This link does not work.

Round 2

Reviewer 1 Report

The necessary changes have been made. As far as I am concerned, the manuscript can be published now.

Reviewer 2 Report

The authors have significantly improved the ms. The research is valuable and worth publishing.